# Sortase A Inhibitor Protein Nanoparticle Formulations Demonstrate Antibacterial Synergy When Combined with Antimicrobial Peptides

**DOI:** 10.3390/molecules28052114

**Published:** 2023-02-24

**Authors:** Sitah Alharthi, Amirali Popat, Zyta Maria Ziora, Peter Michael Moyle

**Affiliations:** 1School of Pharmacy, Pharmacy Australia Centre of Excellence, The University of Queensland, Woolloongabba, QLD 4102, Australia; 2Department of Pharmaceutical Science, School of Pharmacy, Shaqra University, Riyadh 11961, Saudi Arabia; 3Institute for Molecular Bioscience (IMB), The University of Queensland, St Lucia, QLD 4072, Australia

**Keywords:** antimicrobial resistance, 𝛽-lactoglobulin, protein nanoparticles, sortase A inhibitors, synergy

## Abstract

Sortase A (SrtA) is an enzyme which attaches proteins, including virulence factors, to bacterial cell walls. It is a potential target for developing anti-virulence agents against pathogenic and antimicrobial resistant bacteria. This study aimed to engineer 𝛽-lactoglobulin protein nanoparticles (PNPs) for encapsulating safe and inexpensive natural SrtA inhibitors (SrtAIs; *trans*-chalcone (TC), curcumin (CUR), quercetin (QC), and berberine (BR)) to improve their poor aqueous dispersibility, to screen for synergy with antimicrobial peptides (AMPs), and to reduce the cost, dose, and toxicity of AMPs. Minimum inhibitory concentration (MIC), checkerboard synergy, and cell viability assays were performed for SrtAI PNPs against Gram-positive (methicillin-sensitive and -resistant *S. aureus*) and Gram-negative (*E. coli*, *P. aeruginosa*) bacteria alone and combined with leading AMPs (pexiganan, indolicidin, and a mastoparan derivative). Each SrtAI PNP inhibited Gram-positive (MIC: 62.5–125 µg/mL) and Gram-negative (MIC: 31.3–500 µg/mL) bacterial growth. TC PNPs with pexiganan demonstrated synergy against each bacteria, while BR PNPs with pexiganan or indolicidin provided synergy towards *S. aureus*. Each SrtAI PNP inhibited SrtA (IC_50_: 25.0–81.8 µg/mL), and did not affect HEK-293 cell viability at their MIC or optimal synergistic concentrations with AMPs. Overall, this study provides a safe nanoplatform for enhancing antimicrobial synergy to develop treatments for superbug infections.

## 1. Introduction

Antimicrobial-resistant pathogens are an important global health and economic issue. Due to the spread of antibiotic resistant pathogens, current antimicrobial agents are losing their efficacy against important disease-associated microbes, leading to increases in morbidity and mortality [1]. To reduce these problems, the development of drugs that interfere with bacterial virulence factors has received increased attention as an alternative to traditional antibiotics. These anti-virulence therapies inhibit virulence pathways in bacteria, reducing the severity of infections without influencing pathogen viability. This, in turn, is associated with decreased selective pressure for the development of antimicrobial resistance [2].

Sortase A (SrtA) is a membrane-associated cysteine transpeptidase, which is responsible for the attachment of many proteins (including virulence factors) to the cell wall of Gram-positive and some Gram-negative bacteria (e.g., *Colwellia psychrerythraea, Bradyrhizobium japonicum*, *Saccharophagus degradans*, *Shewanella oneidensis* and *Shewanella putrefasciens*) [3,4]. Where proteins contain an LPXTG (X is any amino acid) sorting signal near their carboxyl-terminus, SrtA catalyzes the cleavage of the threonine-glycine bond, followed by covalent attachment of the protein to cell wall peptidoglycan [5].

SrtA is considered to be a useful target for the development of anti-virulence agents [6] since it: (i) is an extracellular membrane-bound enzyme, and therefore readily accessible to anti-virulence compounds, (ii) is not essential for bacterial viability, and (iii) is not present in eukaryotic cells [7,8]. Many naturally derived sortase A inhibitors (SrtAIs), (e.g., *trans*-chalcone (TC), curcumin (CUR), quercetin (QC), and berberine chloride (BR)) demonstrate poor aqueous solubility, which can be improved by the addition of organic solvents. However, the concentrations of organic solvents required to maintain their aqueous solubility is often associated with significant toxicity when administered to eukaryotic cells [9], animals, and humans. Thus, the identification of carrier systems, which improve aqueous solubility and possess low toxicity, is key to developing these compounds as alternative antimicrobial approaches [10].

Nanoparticles (NPs), as a class, possess many advantages for drug delivery. For example: (i) their surface charge [11], morphology, size, and hydrophilicity/hydrophobicity can be modified [12], enabling incorporation and release of various species in a controlled manner; [13,14] (ii) they can improve the solubility and stability of encapsulated compounds; and (iii) they can reduce the toxicity of biologically active components by providing a means to target them to their site of action, and reducing their required dose [15]. These properties, in turn, can open opportunities to re-investigate compounds that have previously been overlooked as drugs due to their poor physicochemical properties. Thus, NPs offer many possibilities to improve the efficacy and reduce adverse effects associated with antimicrobial compounds.

Herein, protein NPs (PNPs) were investigated as a means to improve the delivery properties (e.g., aqueous solubility and antimicrobial activity) of the aforementioned poorly water soluble SrtAIs (BR, CUR, QC, and TC), in place of using organic solvents. These SrtAIs were selected as they are readily available, inexpensive, safe, and widely used as dietary supplements. PNPs have attracted considerable attention due to their low toxicity, biodegradability, high loading efficiency, ease of production, and capacity to readily modify their surface properties and size [13]. In this work, β-lactoglobulin (β-LG; 18.3 kDa) NPs, derived from bovine milk, were selected as they are inexpensive, Generally Regarded as Safe (GRAS), and have proven efficacy for encapsulating, as well as improving the aqueous solubility and bioavailability of hydrophobic compounds [16,17].

A simple desolvation fabrication process (Figure 1), where each SrtAI was individually dissolved in a water miscible organic solvent, and subsequently slowly added to an aqueous solution of β-LG [18], was used to generate PNPs. This process causes β-LG to precipitate, and encapsulate each SrtAI. Following solvent removal, these PNPs were used to investigate their ability to improve each SrtAI’s aqueous solubility. In addition, their ability to improve each SrtAI’s antimicrobial efficacy (assessed using a colorimetric broth microdilution minimum inhibitory concentration (MIC) assay [9]) towards a library of important disease-associated Gram-positive (*Staphylococcus aureus* ATCC25923, and methicillin-resistant *S. aureus*, (MRSA) ATCC43300), and Gram-negative (*Escherichia coli* ATCC25922 and *Pseudomonas aeruginosa* ATCC27853) bacteria, and reduce their toxicity towards mammalian cells in comparison to formulations with organic co-solvents, was investigated. Each β-LG PNP was further characterised with respect to its morphology, size, effects on SrtAI crystallinity, and inhibition potency towards SrtA. Finally, each formulation was assessed for its ability to synergistically improve antimicrobial efficacy when combined with a library of leading antimicrobial peptides (AMPs) (pexiganan (PEX), indolicidin (INDO), and [I5, R8] mastoparan (MASTO; contains an isoleucine and arginine at positions 5 and 8, respectively, which ameliorates its toxicity compared to wildtype mastoparan [19]).

Overall, the development of these PNPs offers the potential to produce novel, biologically compatible and biodegradable formulations to improve the aqueous solubility of poorly water soluble SrtAIs, as well as to ensure their co-delivery to bacteria with other antimicrobial agents, such as AMPs. These characteristics improve the chances of antimicrobial synergy between each component, reducing the required dose, cost, and likelihood of toxicity being observed, and owing to both components possessing novel mechanisms of action, each with low potential for AMR development, the chances of antimicrobial resistance developing is minimised. Thus, this system provides a valuable strategy to reduce the significant costs associated with the clinical use of AMPs, improving their ability to be developed as new antimicrobial agents against a variety of microbial pathogens, including superbugs.

## 2. Results and Discussion

### 2.1. Characterization of SrtAI PNPs

Each SrtAI, formulated in PNPs or alone, was dispersed in water to 1 mg/mL and visually assessed for turbidity (Figure 1). The unformulated SrtAIs demonstrated poor aqueous solubility, with significant turbidity and precipitation observed. In contrast, apart from CUR PNPs, the other PNP-formulations yielded clear solutions under these conditions, which were devoid of visible precipitation. This indicated that the PNP formulations could uniformly disperse each SrtAI in aqueous media, at concentrations that are appropriate for assessing their antimicrobial effects, without the need to add toxic concentrations of organic co-solvents.

The particle size distribution (Z-Average, PDI, and intensity mean hydrodynamic diameter) and surface charge (Zeta-potential) for SrtAI-loaded and unloaded PNP groups were measured by DLS (Figure 2) in PBS. Each SrtAI-loaded PNP group demonstrated a similar Z-average hydrodynamic diameter of approximately 170 nm (Figure 2). The polydispersity index for each group was less than 0.3, with BR (0.283), CUR (0.275) and QC (0.241) PNPs demonstrating more heterogeneity than TC PNPs (0.170). These values were also larger than the Z-average hydrodynamic diameter (156 ± 7.30 nm) and PDI (0.198 ± 0.03) observed for unloaded PNPs (Figure 2). In addition, the SrtAI-loaded and unloaded PNP groups demonstrated similar Zeta-potentials of around −6 mV (Figure 2), which can be attributed to the β-LG nanoparticles, rather than the SrtAIs. Previous studies, which used a different method to produce β-LG nanoparticles, also reported negative Zeta-potentials for β-LG nanoparticles [20,21,22].

The morphology of the SrtAI-loaded and unloaded the β-LG nanoparticles was confirmed by TEM (Figure 2). These images demonstrated approximately spherical shaped particles, with a dark shell and a light core, which is characteristic for β-LG PNPs. The average particle size was determined from TEM images and compared to the DLS data (Figure 2). The DLS results showed that the Z-average diameter ranged from 164 to 176 nm for each SrtAI-loaded PNP.

To characterise the effects of PNP formulations on the crystallinity of each SrtAI, DSC was performed on individual SrtAIs, as well as SrtAI-loaded and empty PNPs. This data (Figure 3) revealed sharp endothermic peaks for TC (57 °C), CUR (177 °C), and QC (317 °C), and a broad endothermic peak for BR (190 °C), which were not present in the empty PNP data and matched literature reported values for the melting point of each SrtAIs crystalline form [23,24,25]. In contrast, these endothermic peaks were no longer observed (CUR, BR, QC), or greatly reduced (TC), for the SrtAI-loaded PNPs. This suggests that the PNP formulations were able to incorporate each SrtAI in an amorphous form, which has been associated with improving their dissolution in aqueous solvents [3].

### 2.2. β -LG PNP SrtAI-Loading Capacity

To determine the loading of each SrtAI into β-LG PNPs, an HPLC method was developed. Concentration versus area under the curve calibration curves (Appendix A) were produced for each SrtAI and used to determine the loading capacity of each SrtAI in their PNP formulations. The percent theoretical loading capacity for SrtAIs in β-LG PNPs was calculated according to the following equation:(1)$$\frac{SrtAI\;mass}{SrtA\;mass+\beta -LG\;mass}\times 100=\frac{10\;\mathrm{mg}}{100\;\mathrm{mg}}\times 100=10\%$$

The HPLC data indicated that the loading capacities for TC, CUR, QC and BR in β-LG PNPs were 8.9, 2.3, 4.1 and 7.9%, respectively. For CUR and QC, low loading capacities were observed. This was hypothesized to be in part due to degradation of these compounds, in addition to the small amount of precipitation observed for the CUR β-LG PNPs, as many peaks in addition to the SrtAI and β-LG PNP peaks were observed in the HPLC data (Appendix A).

### 2.3. Antibacterial Activity of Each SrtAI PNP Formulation

MIC values for each SrtAI-loaded and unloaded PNP formulation were determined against two Gram-positive (MRSA ATCC43300 and *S. aureus* ATCC25923) and two Gram-negative (*E. coli* ATCC25922 and *P. aeruginosa* ATCC27853) bacterial strains using a broth microdilution assay with resazurin colorimetric readout (Table 1). The results indicated that each SrtAI PNP formulation possessed antimicrobial activity against all tested bacteria within the assessed concentration range, while no antimicrobial activity was observed for the unloaded PNPs (Table 1, Appendix A). Of significance, identical MIC values were observed for each SrtAI PNP when assessed in methicillin-sensitive and resistant *S. aureus*, indicating that these formulations were not affected by the antibiotic resistance mechanisms present in the ATCC43300 MRSA strain. In addition, the CUR and QC PNPs exhibited superior antimicrobial activity against all assessed bacteria strains, with 8–16 fold higher potency against Gram-negative strains (*E. coli*, and *P. aeruginosa*), and 2-fold higher potency against Gram-positive (MSSA and MRSA) strains compared to TC and BR PNPs (Table 1). Since the Gram-negative strains do not express SrtA, this indicates that these compounds possess antimicrobial activities that are effective against these reference strains independent of SrtA inhibition activity. Furthermore, comparison of MIC data for SrtAI-loaded β-LG PNPs against the same SrtAI in 5% (*v*/*v*) DMSO [9] demonstrated that the PNP formulation did not alter the MIC values that were observed against each bacterial strain for the TC and BR formulations. In comparison, a 4-fold improvement in potency against *S. aureus*, and an 8-fold improvement against *E. coli* and *P. aeruginosa* was observed with the CUR and QC PNP formulations.

### 2.4. Identification of Synergistic SrtAI PNP Combinations with AMPs

A checkerboard assay was used to analyse for synergistic combinations of SrtAI-loaded PNPs with AMPs (PEX, INDO, MASTO). Synergistic combinations are desirable as they reduce the amount of each component necessary for antimicrobial efficacy, which in turn can reduce the cost, adverse effects, and development of antimicrobial resistance for each component. Each formulation was assessed against Gram-positive (Appendix A, *S. aureus* (MSSA) ATCC25923; Appendix A, MRSA ATCC43300) and Gram-negative (Appendix A, *E. coli* ATCC25922; Appendix A, *P. aeruginosa* ATCC27853) bacteria. From this data, fractional inhibitory concentration index (ΣFIC) values were calculated (≤0.5, synergy; >0.5 to 1, additive; >1 to 4, indifferent; and >4 antagonistic combinations), and summarised in Table 2. An example of a synergistic combination, TC-loaded PNPs combined with PEX, against *S. aureus* (MSSA) ATCC25923, MRSA ATCC43300, *E. coli* ATCC25922, and *P. aeruginosa* ATCC27853, is presented in Appendix A.

From the data summarised in Table 2, it can be observed that TC PNPs, when combined with PEX, demonstrated synergy against all assessed bacteria strains. Representative data for MSSA, MRSA, *E. coli*, and *P. aeruginosa* (Appendix A) have been provided. The combination of TC PNPs with PEX reduced the MIC of TC PNPs by 4-fold against *E.coli* and *P. aeruginosa*, and 8-fold against MSSA and MRSA, and decreased the MIC of PEX by 8-fold against *E. coli* and 4-fold against *P. aeruginosa*, MSSA, and MRSA. In comparison, additive antimicrobial activity was observed against each bacteria strain when INDO or MASTO were combined with TC PNPs (Appendix A).

BR PNPs also displayed synergy against MSSA and MRSA when delivered with PEX, similar to TC PNPs, and additionally with INDO (Appendix A), while additive antimicrobial activity was observed for all other BR PNP combinations with AMPs (Appendix A). Thus, TC PNPs, when combined with PEX, displayed superior synergy against Gram-negative bacteria strains compared to BR PNPs. The combination of BR PNPs with PEX reduced the MIC of BR PNPs by 8- and 16-fold, respectively, and the MIC of PEX by 4-fold against MSSA and MRSA.

In the case of CUR or QC PNPs, no synergistic combinations with AMPs were identified (Table 2). However, CUR PNPs demonstrated additive antimicrobial activity for each combination, with the exception of PEX against MSSA and *E. coli*, which were classified as indifferent. Similarly, QC PNPs demonstrated additive antimicrobial activity for each combination, with the exception of INDO against MSSA and MRSA, and MASTO against *E. coli* and *P. aeruginosa*, which were also classified as indifferent (Table 2).

### 2.5. Assessment of SrtA Inhibition

SrtA inhibition has been widely investigated as a means to improve the sensitivity of bacteria to host immune responses, as well as to increase the potency of co-delivered antibiotics and overcome antibiotic resistance [9]. Previous research has investigated the SrtA inhibition potency of the SrtAIs used herein, when administered using 5% *v*/*v* DMSO to improve their solubility [9]. This concentration of DMSO, however, is generally a maximum accepted concentration for enzyme assays, and is likely to affect enzyme structure, selectivity and activity, with significantly lower concentrations (e.g., 0.1% *v*/*v*) usually preferred. Because the PNP formulations of each SrtAI can be directly dispersed in aqueous buffers, without the need for organic solvents to improve their solubility, an aim of this work was to assess how the inhibition of SrtA with these formulations compared to the data previously acquired in the presence of 5% (*v*/*v*) DMSO. For this purpose, a FRET-based assay was used to quantify the progress of SrtA_ΔN59_-catalyzed cleavage of a quenched fluorescent peptide Abz-LPETGK(Dnp)-NH_2_. Upon cleavage, quenching of the 2-aminobenzoyl (Abz) fluorophore by 2,4-dinitrophenyl (DNP) is lost, allowing for monitoring of the reaction using the 335/420 nm excitation/emission wavelengths.

To determine the SrtA inhibition potency for each formulation, kinetic readouts of Abz-LPETGK(Dnp)-NH_2_ cleavage by the SrtA_ΔN59_ enzyme were recorded over 60 min. From this data, dose-response curves were generated and IC_50_ values calculated (Figure 4). These data demonstrated that the SrtA_ΔN59_ inhibition potency for BR, CUR, and TC PNPs were similar (IC_50_ 25–35 µg/mL, with overlapping 95% confidence intervals), while QC PNPs were approximately 2–3-fold less potent (IC_50_ 81.9 µg/mL). This data for TC and CUR PNPs were equivalent to what was observed for these SrtAIs when administered in 5% (*v*/*v*) DMSO [8]. In comparison, BR was approximately 5-fold more potent (IC_50_ 6.9 µg/mL), and QC 2-fold less potent (42.2 µg/mL), when administered in 5% (*v*/*v*) DMSO compared to the PNP formulations. The similar IC_50_ values observed for these PNP formulations compared to the 5% (*v*/*v*) DMSO formulations suggest that the SrtA inhibition that was observed in the presence of 5% (*v*/*v*) DMSO could be attributed to the SrtAIs themselves for CUR, QC, and TC. In comparison, the BR PNP formulation was less potent compared to the 5% (*v*/*v*) DMSO formulation. Since each SrtAI needs to be released from the PNP formulation in order to inhibit SrtA, this may indicate that BR is not as readily released from the PNPs compared to the other formulations, or that it aggregates upon release from these formulations, reducing its ability to interact with, and inhibit SrtA. Finally, comparison of the IC_50_ values that were calculated for each SrtAI PNP formulation (Figure 4) to the MIC data in Table 1, indicated that all MIC values were higher than the IC_50_ values, with the exception of the QC PNP group, and thus significant inhibition of the bacterial SrtA enzyme was likely experienced at the MIC value.

### 2.6. Cell Viability

In a previous study from the investigators, DMSO was used to improve the solubility of each SrtAI investigated herein, which led to significant reductions in HEK-293 cell viability [9]. In particular, no viable cells were observed in TC (1000–31.3 μg/mL) or CUR (2000–62.5 μg/mL) treated groups across the assessed concentration ranges, which included their MIC values. In comparison, BR (1000–31.3 μg/mL) and QC (2000–62.5 μg/mL) retained cell viability at lower concentrations, but demonstrated greatly reduced cell viability at 250 μg/mL and higher concentrations. Because the PNP formulations, developed herein, allow for the delivery of each SrtAI without the need for DMSO to maintain their solubility, investigations into the effects of these formulations on HEK-293 cell viability alone, and in combination with the AMPs (PEX, INDO, and MASTO), were performed, to determine if PNP formulations demonstrated reduced toxicity in comparison to the DMSO formulations, while maintaining their antimicrobial and SrtAI activity. For this purpose, a resazurin-based cell viability assay was used, with the concentration ranges for each SrtAI PNP or for the synergistic AMP/SrtAI PNP combinations (PEX/TC, PEX/BR, and INDO/BR PNPs; Table 2) in the two-fold dilution series selected to ensure three concentrations above their SrtAI PNP MIC value against *S. aureus* (Table 1) or the optimal combination against *S. aureus* as determined by the checkerboard assay (ΣFIC) (Table 2), respectively, as well as two concentrations below these values.

The results of this study indicated that each SrtAI PNP formulation (Figure 5), or synergistic AMP/SrtAI PNP combination (Figure 6), did not affect cell viability at their MIC or optimal combinations (ΣFIC). In addition, the CUR and QC PNPs (Figure 5), as well as the INDO/BR and PEX/BR PNP combinations (Figure 6), demonstrated minimal effects (< 20% reduction) on cell viability at concentrations up to 8-fold higher than their MIC or optimal combinations (ΣFIC), respectively. In comparison, BR PNPs demonstrated a small reduction in cell viability with a 23% and a 27% reduction at concentrations that were 4- (500 µg/mL) and 8-fold (1000 µg/mL) higher than their MIC values (Figure 5). Finally, TC PNPs demonstrated minimal effects on cell viability at a concentration 2-fold (125 µg/mL) higher than its MIC, while at 4-fold (500 µg/mL) and higher concentrations, cell viability was reduced to less than 20% (Figure 5). In comparison, the PEX/TC PNP combination appeared to have less effect on cell viability, with the combination at a 4-fold higher concentration (6.25 μg/mL PEX + 62.5 μg/mL TC PNPs) than its optimal combination (ΣFIC) demonstrating minimal (<20% reduction) effects on cell viability, and cell viability remaining at approximately 50% when the combination was used at 8-fold higher concentration (12.5 μg/mL PEX + 125 μg/mL TC PNPs) than its optimal combination (ΣFIC) (Figure 6). This suggests that the synergistic combinations can not only improve antimicrobial efficacy, but may also reduce toxicity, improving the capacity to translate these components for future clinical use.

## 3. Materials and Methods

### 3.1. Materials

𝛽-LG, tris (hydroxymethyl) aminomethane (Tris) hydrochloride, TC, QC, BR, CUR, resazurin, agar, methanol (MeOH), ethanol (EtOH), sodium dodecyl sulfate (SDS), hydrochloric acid (HCl), Mueller—Hinton broth (MHB), and dimethylsulfoxide (DMSO) were purchased from Merck (Castle Hill, NSW, Australia). Propylene glycol liquid (USP) was purchased from the Melbourne Food Ingredient Depot (Melbourne, VIC, Australia). Triton X-100, CaCl_2_ and NaCl were purchased from Chem-Supply (Gillman, SA, Australia). Triisopropylsilane (TIPS) was purchased from Alfa Aesar (Tewksbury, MA, USA). The SrtA_ΔN59_ enzyme was expressed and purified as previously described [4]. Dulbecco’s modified eagle medium (DMEM), fetal bovine serum (FBS), and penicillin/streptomycin were purchased from Thermo Fisher Scientific (Scoresby, VIC, Australia). PEX, INDO, MASTO, the quenched fluorescent resonance energy transfer (FRET) peptide (Abz-LPETGK(Dnp)-NH_2_), and Gly_3_-amide peptides were synthesized as previously described [9]. The bacterial library used herein consisted of *S. aureus* (ATCC25923), MRSA (ATCC43300), *E. coli* (ATCC25922), and *P. aeruginosa* (ATCC27853).

### 3.2. Equipment

A Biotage^®^ Initiator^+^ Alstra microwave peptide synthesizer (Uppsala, Sweden) was used for microwave-assisted peptide synthesis. Liquid chromatography-mass spectrometry (LC-MS) was performed on a Shimadzu LCMS-2020 system, with LabSolutions 5.89 software (Kyoto, Japan). Analytical RP-HPLC was performed on an Agilent 1200 Series binary LC system (Santa Clara, CA, USA) (G1379B degasser, G1312A binary pump, G1329A ALS autosampler, G1316A thermostatted column compartment, G1315D diode array detector and Chemstation Rev.B.04.02 software). Separations were conducted on a C18 stationary phase, for which a Grace Vydac C18 analytical (Columbia, MD, USA) (218TP54; 150 × 4.6 mm; 5 μm) column was used with linear gradients of Solvent A (0.1% (*v*/*v*) trifluoroacetic acid (TFA)-water) and solvent B (90% (*v*/*v*) acetonitrile (MeCN)/0.1% (*v*/*v*) TFA-water) at 1 mL/min, and UV absorbance detection at 214 nm. For fluorescence measurements, an Envision Multilabel Plate Reader (Perkin Elmer, Rowville, VIC, Australia) was used. Sonication was performed using a Branson (Brookfield, CT, USA) CPX2800H-E ultrasonic bath. Thermogravimetric analysis (TGA) and differential scanning calorimetry (DSC) were performed on a Mettler Toledo (Columbus, OH, USA) TGA/DSC 2 STAR instrument. Dynamic light scattering (DLS) measurements of particle size and Zeta-potential were performed using a Zetasizer Nano ZS (Malvern Panalytical, Malvern, UK) instrument. Transmission electron microscopy (TEM) was performed on a Hitachi (Ibaraki, Japan) HT7700B at 100 kV. Fourier-transform infrared spectroscopy (FTIR) was performed on a Perkin-Elmer (Liantrisant, UK) Spectrum TWO instrument. Ultrapure (18 MΩ) deionized reverse osmosis water was prepared using a Millipore (Burlington, MA, USA) Simplicity UV ultrapure water system.

### 3.3. Preparation of SrtAI-Loaded PNPs

SrtAI PNPs were prepared according to an established method [18] with modifications. Briefly, 𝛽-LG (90 mg) and each SrtAI (10 mg; TC, CUR, QC, and BR) were individually dissolved (to 1 mg/mL) in ultrapure water or MeOH, respectively, and stirred (700 RPM, 30 min, 37℃) in the dark. Each SrtAI/MeOH stock solution, or MeOH (1 mL) alone (for the blank PNP control), was then added in a dropwise manner to individual 𝛽-LG aqueous stock solutions over ~15 min, and left to stir (700 RPM, 30 min, 37℃) in the dark overnight. The MeOH was then removed by rotary evaporation (40 ℃, 40 mbar, 20 min), and the mixtures snap frozen in liquid nitrogen for 10 min before freeze drying to yield the SrtAI/PNP complexes (1:9 *w/w*) or blank PNP controls.

### 3.4. Characterization of SrtAI-Loaded PNPs

The morphology and size of each PNP was investigated by TEM at 100 kV. For this purpose, PNPs were dispersed in ultrapure water (to 1 mg/mL), gently bath sonicated (3 min), and diluted to 100 µg/mL with ultrapure water. Nanoparticle suspension droplets were individually air dried on carbon-coated copper grids (formvar/heavy carbon-coated, 300 mesh), stained with 2% *w*/*v* uranyl acetate in water, and imaged. The intensity weighted mean hydrodynamic diameter (Z-Ave), polydispersity index (PDI), intensity mean diameter of the main peak, and Zeta-potential were measured by DLS. For this purpose, individual SrtAI PNP samples, or blank PNPs, were prepared to 1 mg/mL in PBS, added to disposable folded capillary cells, and measured at 25 ℃. TGA and DSC were performed to investigate the thermal behaviour of each SrtAI-loaded PNP. For these experiments, freeze-dried samples of SrtAIs, SrtAI PNPs or blank PNPs (5 mg) were individually added to a 70 µL alumina crucibles and heated over a temperature range of 50 to 900 ℃, at 10 ℃/min with 20 mL/min airflow. Interactions between individual SrtAIs and 𝛽-LG PNPs were investigated using FTIR. For this purpose, each sample (2 mg) was mixed with KBr (1:100 ratio), and the FTIR spectra recorded over a 4000 to 400 cm^−1^ wavenumber range at RT.

### 3.5. Loading Capacity Determination

To determine the loading capacity of the prepared formulations, 1 mg of the SrtAI PNPs was dissolved in 1 mL of 45% (*v*/*v*) MeCN/0.1% (*v*/*v*) TFA-water, centrifuged (21.3 k ×*g*, 30 min), and analysed by RP-HPLC [26]. Area under the curve (AUC) data for the SrtAI and 𝛽-LG components were measured and compared to standard curves to quantify the amount of each component in the formulation.

The theoretical loading capacity was calculated using the following equation [19]:(2)$$Theoretical\;Loading\;capacity\;\left(\%\right)=\frac{SrtAI\;weight}{SrtAI\;weight+NP\;weight}\times 100$$

### 3.6. Preparation of Bacterial Inoculums

Bacterial inoculums were prepared according to the Clinical and Laboratory Standards Institute (CLSI) guidelines [27]. From individual overnight *S. aureus* ATCC25923, MRSA ATCC43300, *E. coli* ATCC25922, or *P. aeruginosa* ATCC27853 agar plate cultures, two-to-three individual colonies were mixed in sterile normal saline (2 mL), followed by adjustment of turbidity to the 0.5 McFarland standard (1 × 10^8^ colony-forming unit (CFU)/mL). A sample of this suspension (100 µL) was then diluted 100-fold with MHB to yield 1 × 10^6^ CFU/mL bacterial stocks for broth microdilution and checkerboard assays.

### 3.7. Broth Microdilution Assays

To investigate the antimicrobial activity of SrtAI PNPs or AMPs alone, their MICs were determined against the bacterial library based on the Clinical and Laboratory Standards Institute (CLSI) BMD method with modifications [28]. Stock solutions of SrtAI PNPs were individually prepared in MHB (1 mL) to 1 mg/mL final SrtAI concentrations. AMP stock solutions were individually prepared in 0.01% (*v*/*v*) DMSO-MHB to 100 (PEX), 300 (INDO), and 200 (MASTO) µg/mL. Subsequently, SrtAI PNP and AMP stock mixtures (200 µL/well) were added to individual column 1 wells in a sterile 96-well round-bottomed microplate. MHB (100 µL/well) was then dispensed into columns 2 to 11, and 200 µL/well into column 12. A 2-fold serial dilution of the SrtAI PNPs and AMPs in column 1 was then prepared by pipetting 100 µL from column 1, mixing this with the next column to the right, and repeating this process until column 10, with 100 µL of the column 10 contents subsequently discarded. A bacterial inoculum (100 µL/well; 1 × 10^6^ CFU/mL), corresponding to a single bacterial library strain per plate, was then added to columns 1 to 11. The plate was incubated for 24 h at 37 ℃, and the turbidity of the wells visually assessed to determine the MIC. To provide an additional colorimetric MIC readout, 15 mg/mL resazurin in water [28] (30 µL/well) was added to each well, and the plate incubated for 2 h at 37 ℃ [29]. The MIC was defined as the lowest concentration that displays no visible turbidity and retains a dark blue color. Column 11 (+) contained no SrtAI PNP or AMP and provided a bacterial growth control. Column 12 (−) contained only MHB and provided a sterility control. The experiment was performed in triplicate with three independent experiments for each SrtAI PNP and AMP.

### 3.8. Checkerboard Assay

A checkerboard assay was performed to investigate antimicrobial synergies between combinations of SrtAI PNPs and AMPs towards the bacterial library using an established protocol [30]. Briefly, SrtAI PNP stocks (containing 2 mg/mL of individual SrtAIs) were prepared in MHB and 2-fold serial diluted with MHB to yield SrtAI concentrations between 2 mg/mL and 31.3 µg/mL. Similarly, individual PEX (200 µg/mL), INDO (600 µg/mL) and MASTO (400 µg/mL) stocks were prepared in 0.01% (*v*/*v*) DMSO-MHB, and 2-fold serial diluted with 0.01% (*v*/*v*) DMSO-MHB to 3.2, 9.4, and 6.3 µg/mL final AMP concentrations, respectively. The individual AMP stock serial dilutions were dispensed (50 µL/well) into column 1 to 7 wells, from highest to lowest stock concentrations, with 0.1% (*v*/*v*) DMSO-MHB (50 µL/well) dispensed into column 8 wells. Subsequently, individual SrtAI PNP stock serial dilutions were dispensed (50 µL/well) into rows A to G, from highest to lowest stock concentrations, with MHB (50 µL/well) dispensed into row H. A bacterial inoculum (100 µL/well; 1 × 10^6^ CFU/mL), corresponding to a single bacterial library strain per plate, was then added into each well, and the plates were incubated for 24 h at 37 ℃. Subsequently, 15 mg/mL resazurin in water (30 µL/well) was added to each well, and the plate incubated for 2 h at 37 ℃. MIC values were defined as the lowest concentration that remained dark blue in column 8 for SrtAIs and row H for AMPs.

To determine synergistic combinations, the fractional inhibitory concentration index (ΣFIC) was calculated according to the following formula:(3)$$\Sigma FIC=\frac{MIC\;of\;AMP\;in\;combination\;with\;SrtAI}{MIC\;of\;AMP\;alone}+\frac{MIC\;of\;SrtAI\;PNP\;in\;combination\;with\;AMP}{MIC\;of\;SrtAI\;PNP\;alone}$$

For calculating the ΣFIC values, wells that demonstrated the largest decrease in MIC values for one of the components were selected. The combinations were defined as synergistic (ΣFIC ≤ 0.5), additive (ΣFIC between >0.5 to 1), indifferent (ΣFIC between >1 to 4), or antagonistic (ΣFIC > 4) on the basis of ΣFIC values.

### 3.9. SrtA Inhibition Assay

SrtA_ΔN59_ inhibition was assessed in Costar black 96-well half-area, flat-bottomed microplates as previously described [9]. Assay wells (150 µL/well) were assembled to include a reaction buffer (to 50 mM Tris pH 7.5, 150 mM NaCl, 5 mM CaCl_2_, 0.01% (*v*/*v*) DMSO, and 0.1% (*v*/*v*) Triton X-100 final concentrations), 4 µM SrtA_ΔN59_, half-log serial dilutions of each SrtAI PNP (from 500 to 1.6 µg/mL final SrtAI concentrations), 100 µM Gly_3_-amide peptide, and 10 µM Abz-LPETGK(Dnp)-NH_2_ quenched fluorescent FRET peptide. Each component was added in the order listed above. The plate was incubated at 37 ℃ with shaking for 1 h, with the cleaved FRET peptide fluorescence measured at 335/420 nm (ex/em) every 5 min. Wells without SrtA_ΔN59_ were used to measure the background fluorescence. Wells without SrtAI PNPs were used as a negative control.

### 3.10. Cell Viability

HEK-293 cell viability was measured in the presence of SrtAI PNPs alone, or in combination with AMPs, using a resazurin reduction assay [31]. HEK-293 cells were seeded (5 × 10^3^ cells/well) into black 96-well plates containing complete media (DMEM, 10% (*v*/*v*) FBS and 1% (*v*/*v*) penicillin/streptomycin; 100 μL) and incubated (37 ℃, 5% CO_2_, 48 h) to 80% confluency. The media was then discarded, and the cells treated with complete media/0.01% (*v*/*v*) DMSO (100 μL/well) containing a two-fold dilution series of SrtAI PNPs, or SrtAI PNPs in combination with AMPs. The concentrations that were investigated ensured that two concentrations above the MIC, and three below, were assessed. Positive and negative toxicity controls consisted of cells treated with 10% (*v*/*v*) SDS containing 0.1 M HCl or complete media/0.01% (*v*/*v*) DMSO, respectively. Empty wells treated with complete media/0.01% (*v*/*v*) DMSO were used as a background control. The plates were then incubated (37 °C, 5% CO_2_, 24 h), the supernatant removed, and 100 μM resazurin in PBS (50 μL) added to each well. The plates were then incubated (37 °C 5% CO_2_, 4 h) and resorufin fluorescence (F) was read (560/590 nm ex/em).

The percentage of viable cells was calculated according to the following formula:(4)$$Cell\;viability\;\left(\%\right)=\frac{Fluorescence_{sample}-Fluorescence_{background}}{Fluorescence_{negative\;control\;}-Fluorescence_{background}}\times 100$$

## 4. Conclusions

Prior to the identification of antibiotics, severe infections were often fatal. The development of antimicrobial-resistant pathogens threatens to send us back to a time without effective antibiotics. Thus, the development of innovative antimicrobial agents, with novel mechanisms of action, against which the development of antimicrobial resistance is unlikely, is an area of major public health importance. The work herein demonstrates one such strategy, based on inexpensive, natural antimicrobial agents (TC, CUR, QC, and BR) that are readily available as dietary supplements, and which inhibit bacterial SrtA. This enzyme is necessary for the attachment of virulence factors to the cell wall of Gram-positive and some Gram-negative bacteria. Hence, its inhibition reduces the capacity for bacteria to cause disease, and increases the sensitivity of bacteria to other antibiotics. However, these agents exhibit poor aqueous solubility, which hinders their bioavailability, potential routes of administration, and capacity to be marketed as pharmaceutical products. As a simple, inexpensive, and safe means to improve the aqueous dispersibility of these compounds, the capacity to encapsulate each SrtAI within 𝛽-LG PNPs was investigated. This process formed sub-200 nm particles, which dispersed readily in aqueous media, in comparison to the suspensions that were observed with non-formulated SrtAIs. These formulations demonstrated antimicrobial activity against Gram-positive (*S. aureus*) and Gram-negative (*E. coli* and *P. aeruginosa*) bacteria, including MRSA, with identical (TC and BR PNPs) or improved (CUR and QC PNPs) potency compared to their delivery with 5% (*v*/*v*) DMSO. The formulations were also demonstrated to effectively inhibit SrtA, and to possess, in most cases, significantly reduced effects on HEK-293 cell viability compared to 5% (*v*/*v*) DMSO formulations.

The capacity for SrtAIs to synergise with, and improve the activity of, AMPs while reducing their toxicity, required dose, and cost was also a focus herein. The results identified that TC PNPs when combined with PEX yielded synergistic improvements in antimicrobial activity towards each Gram-positive and Gram-negative bacteria in the assessed library, while BR PNPs combined with PEX or INDO provided synergy only in the case of Gram-positive bacteria. These synergistic combinations also reduced the dose of each component required for antimicrobial activity, and in turn significantly reduced their effects on HEK-293 cell viability over the assessed concentration ranges. This data provided guidance around combinations of SrtAIs and AMPs that could be used to take advantage of synergies, and thus increases the capacity to generate new combination formulations for developing AMPs as marketable products. Overall, the findings from this study demonstrate that 𝛽-LG can be considered as a useful platform to develop PNPs for the delivery of the SrtAIs assessed herein to improve their aqueous solubility and antimicrobial effects, and that these formulations may assist in improving the antimicrobial activity of AMPs, as well as reducing their toxicities and cost.

## Data Availability

Data are covered within the article or Appendix A.

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
