# Peer review of "Sortase A Inhibitor Protein Nanoparticle Formulations Demonstrate Antibacterial Synergy When Combined with Antimicrobial Peptides"

_molecules, 2023, doi:10.3390/molecules28052114_

Round 1

Reviewer 1 Report

The author developed a series of SrtAI protein nanoparticles to improve the poor aqueous solubility and study the synergy with antimicrobial peptides. Protein nanoparticles could significantly increase the solubility in aqueous solution, and the nanoparticles exhibited desired size distribution and morphology. Minimum inhibitory concentration (MIC), checkerboard synergy, and cell viability assays demonstrated that synergistic combinations can not only improve antimicrobial efficacy, but may also reduce toxicity. In general, it is an appealing work with innovation and guiding significance in the pharmaceutical field. However, the following comments need to be addressed before consideration for publication in Molecule.

1. In Figure 1, the images of unformulated and PNP-formulated SrtAI in water are blurred, clearer pictures should be provided.

2. In Figure 2, although the bar in the TEM images represents 2 nm, the length of bars is not unified. So, it would be better to unify the magnifications of each TEM images.

3. In the Page 9, line 318, this sentence “This was hypothesised to be in part due to degradation of these compounds, as many peaks in addition to SrtAI and β-LG PNP peaks were observed in the HPLC data (Figure S2).” indicates that the reason for low loading capacities of CUR and QC is to be easily degraded. However, CUR and QC are small molecule, they are stable in aqueous solution. Meanwhile, minor precipitation in CUR PNPs aqueous solution could be observed, which may be the reason for the low loading capacity. So, the reason for low loading capacities of CUR and QC needs to be further explored. Or, some references need to be provided to support the “degradation hypothesis”.

4. In supporting information, Figures S3 and S4 are not provided.

5. In page 14, line 423, it is said that “Since each 423 SrtAI needs to be released from the PNP formulation in order to inhibit SrtA, this may 424 indicate that BR is not as readily released from the PNPs compared to the other formula-425 tions, or that it aggregates upon release from these formulations, reducing its ability to 426 interact with, and inhibit SrtA.” It is suggested to conduct releasing assay to investigate the releasing profile of SrtAI PNPs.

6. there are some grammar and style problems in this paper, for example: in page 1 line 42, a full stop is lacked before the sentence “Where proteins contain an LPXTG (where X is any amino acid) sorting signal near their carboxyl-terminus, SrtA catalyzes the cleavage of the threonine-glycine bond, followed by covalent attachment of the protein to cell wall peptidoglycan.”; in page 10 line 331, this sentence “Thus, making these formulations potentially useful treatments for this important cause of severe infectious disease” lacks verb; in Table 2, the “CUR PNPs” should be arranged in the same row, like other “PNPs”.

Author Response

Reviewer 1 

Comments and Suggestions for Authors

  1. In Figure 1, the images of unformulated and PNP-formulated SrtAI in water are blurred, clearer pictures should be provided.

Response:  Thank you for alerting us to this. The version of the manuscript that was submitted has good quality figures in it. We suspect that the copy that was supplied for review might have the image resolution reduced in it. Despite this, we have found the original images and created new versions of this image which meet the journals image formatting requirements and inserted this into the manuscript.

  1. In Figure 2,although the bar in the TEM images represents 2 nm, the length of bars is not unified. So, it would be better to unify the magnifications of each TEM images. 

Response:  We agree that it is easier for readers to compare these TEM images if the scale bars are the same for each figure. To address this, we have resized all of the TEM panels in Figure 2 so the scale bar is 2 microns.

  1. In the Page 9, line 318, this sentence “This was hypothesised to be in part due to degradation of these compounds, as many peaks in addition to SrtAI andβ-LG PNP peaks were observed in the HPLC data (Figure S2).” indicates that the reason for low loading capacities of CUR and QC is to be easily degraded. However, CUR and QC are small molecule, they are stable in aqueous solution. Meanwhile, minor precipitation in CUR PNPs aqueous solution could be observed, which may be the reason for the low loading capacity. So, the reason for low loading capacities of CUR and QC needs to be further explored. Or, some references need to be provided to support the “degradation hypothesis”. 

Response:  HPLC data was provided in the supporting information (Figure S2) which demonstrates the presence of peaks in addition to the beta-lactoglobulin peak (~ 17.8 min) and the SrtAI peaks (the other major peaks in the chromatograms). This was particularly true for the curcumin PNPs, with some additional peaks seen for the QC PNPs, with the figures cited in the manuscript. The pure starting materials did not contain these peaks. The HPLC solvent conditions that were used ensured the solubilization of each of the small molecule sortase A inhibitors that were used, and as such precipitation cannot be the reason for these peaks as suggested by the reviewer.

Curcumin and quercetin are well known to be chemically unstable at neutral to alkaline pH – many references (as well as our personal experience) support this. For example:

Kharat, M., Du, Z., Zhang, G., & McClements, D. J. (2017). Physical and chemical stability of curcumin in aqueous solutions and emulsions: Impact of pH, temperature, and molecular environment. Journal of agricultural and food chemistry, 65(8), 1525-1532.

  • ‘Pure curcumin was highly unstable to chemical degradation in alkaline aqueous solution (pH ≥ 7.0)’
  • ‘At physiological pH, curcumin rapidly degrades to bicyclopentadione through autooxidation, with cleavage products such as bicyclopentadione, vanillin, and ferulic acid being formed’

Nimiya Y, Wang W, Du Z, Sukamtoh E, Zhu J, Decker E, Zhang G. Redux modulation of curcumin stability: redox active antioxidants increase chemical stability of curcumin. Molecular Nutrition and Food Research 2016;60:487-94

  • Found that curcumin has poor stability in aqueous buffer at physiological pH.
  • HPLC analysis of curcumin in PBS indicated 80-90% degradation in ~ 12 min at room temperature.

Some references for poor chemical stability of quercetin are provided below:

Wang, W., Sun, C., Mao, L., Ma, P., Liu, F., Yang, J., & Gao, Y. (2016). The biological activities, chemical stability, metabolism and delivery systems of quercetin: A review. Trends in Food Science & Technology, 56, 21-38.

  • Quercetin is poorly stable at pH > 7. Subject to oxidation and other degradation pathways.

Gao L, Liu G, Wang X, Liu F, Xu Y, Ma J. Preparation of a chemically stable quercetin formulation using nanosuspension technology. International Journal of Pharmaceutics 2011;404:231-37

  • Demonstrated a 28.3% reduction in quercetin content and obvious discolouration over a 1 month period at 25 C no light.

We have added that the curcumin PNPs group also demonstrated a small amount of precipitation (Figure 1), which may contribute to the low loading capacity for CUR PNPs by adding‘ in addition to the small amount of precipitation observed for the CUR b-LG PNPs’ to the sentence.

  1. In supporting information, Figures S3 and S4 are not provided.

Response: There was an error with the figure numbers in the supporting information that was submitted where the numbers skipped from S2 to S5. We have fixed the figure numbers by changing figures S5, S6, S7, and S8 to figures S3, S4, S5, and S6.

  1. In page 14, line 423, it is said that “Since each SrtAI needs to be released from the PNP formulation in order to inhibit SrtA, this may indicate that BR is not as readily released from the PNPs compared to the other formulations, or that it aggregates upon release from these formulations, reducing its ability to interact with, and inhibit SrtA.” It is suggested to conduct releasing assay to investigate the releasing profile of SrtAI PNPs.

Response:  This study was used as a preliminary screening approach to determine which SrtAI protein nanoparticles were most promising with respect to activity and improved dispersibility in aqueous conditions (as required for possible future in vivo assessment), as well as which formulations demonstrated synergy with antimicrobial peptides. Based on synergy with antimicrobial peptides, the TC protein nanoparticles and BR nanoparticles were the best. This activity was despite the reduction in the berberine IC50 against the SrtA enzyme compared to berberine in 5% DMSO (20.2 mcg/mL vs 6.9 mcg/mL). This reduction however is still in the same order of magnitude and does not result in any significant difference in IC50 values compared to the other SrtAIs used in the current paper (TC, BR and CUR all gave results of 20 mcg/mL). Therefore, there is no need to conduct a release assay at this point. The potential for sustained release from these formulations, in addition to the capacity to utilize coatings to affect the stability and release rate of these compounds is beyond the scope of this paper and is an area of future research interest of our laboratories.

  1. there are some grammar and style problems in this paper, for example: in page 1 line 42, a full stop is lacked before the sentence “Where proteins contain an LPXTG (where X is any amino acid) sorting signal neartheir carboxyl-terminus, SrtA catalyzes the cleavage of the threonine-glycine bond, followed by covalent attachment of the protein to cell wall peptidoglycan.”; in page 10 line 331, this sentence “Thus, making these formulations potentially useful treatments for this important cause of severe infectious disease” lacks verb; in Table 2, the “CUR PNPs” should be arranged in the same row, like other “PNPs”. 

Response: To address this comment,  the manuscript has been revised and the following mistakes have been fixed:

  1. A full stop has been added before the sentence “Where proteins contain an LPXTG (where X is any amino acid) sorting signal…” (Page 1 Line 42)
  2. We disagree with the comment that the sentence “Thus, making these formulations potentially useful treatments for this important cause of severe infectious disease” requires a verb. This sentence is clearly linked to the previous sentence, and indicates that the formulations work against both methicillin-sensitive and -resistance aureus.
  3. Table 2 (page 12) has been adjusted to ensure that ‘CUR PNPs’ is displayed on a single line, instead of spilling over onto two lines as in the original submission.

Reviewer 2 Report

The authors present an interesting study of forming scaffolds by lyophilization. The paper shows new insights into the development of smart materials. However, authors are encouraged to address the following issues before the paper can be published:

1. All equations must be reformatted to the proper font style and size.

2. The methodology of freeze-drying is not clear; in this regard, it is known that immersion of the solution/colloid into liquid nitrogen induces different freezing mass transfer pathways. Depending on the rate of immersion, and time of exposure to the liquid nitrogen, different solvent-freezing pathways induce different nucleation rates that will produce different morphologies, i.e different porosities. Authors must address this issue and offer a plausible scenario that correlates with the functionality of the scaffold.

3. Figure 2 is misleading; particle size distributions are about the same. However, the TEM images show different average particle sizes and distributions. Please clarify these issues.

4. In Figure 3, DSC traces are reported. What are the transitions observed? Can you detect glass transitions, melting points, and degradations?

5. Figure 6 shows extremely large SDs; please elaborate on this issue and its relationship to the type of measurement.

6.

Author Response

Reviewer 2

Comments and Suggestions for Authors

The authors present an interesting study of forming scaffolds by lyophilization. The paper shows new insights into the development of smart materials. However, authors are encouraged to address the following issues before the paper can be published:

  1. All equations must be reformatted to the proper font style and size.

Response: Equations have been embedded using Microsoft Equation Editor in the manuscript to allow editing as per the instructions for authors. The MDPI equations style has been set.

  1. The methodology of freeze-drying is not clear; in this regard, it is known that immersion of the solution/colloid into liquid nitrogen induces different freezing mass transfer pathways. Depending on the rate of immersion, and time of exposure to the liquid nitrogen, different solvent-freezing pathways induce different nucleation rates that will produce different morphologies, i.e different porosities. Authors must address this issue and offer a plausible scenario that correlates with the functionality of the scaffold.

Response: The process that was used to produce the protein nanoparticles was a previously established technique (published as Pujara N, et al. Molecular Pharmaceutics 2021;18:627). Other groups have produced such protein nanoparticles using beta-lactoglobulin. We have made several batches of these particles for this work, with repeatable characterization data observed.

We snap froze the samples i.e. this indicates that they were rapidly immersed in liquid nitrogen, and then left them to sit in the liquid nitrogen for 10 minutes. We have thus added ‘for 10 min’ into section 2.3 to be more specific.

  1. Figure 2 is misleading; particle size distributions are about the same. However, the TEM images show different average particle sizes and distributions. Please clarify these issues.

Response: The distributions are slightly different between each group. An inspection of the range of each distribution in the DLS data shows that there are differences in the particle size ranges that are observed. The TEM data has been adjusted to normalise the scale bar between the groups (requested by another reviewer). This enables easier visual examination between the groups – the previous data had wildly different scale bar sizes, which may have been confusing. All of the loaded protein nanoparticle distributions look quite similar in the TEM data. In addition, it is normal to see differences between the TEM and DLS data as they measure sizing in different ways (e.g. DLS measures bulk data, and includes hydration shells, while TEM includes drying onto the grid, which can affect particle size).

  1. In Figure 3, DSC traces are reported. What are the transitions observed? Can you detect glass transitions, melting points, and degradations?

Response: the peaks that are observed for this work, and described in the results and discussion on page 9 are the melting endotherms for the crystalline forms. While references for these were provided in the manuscript, we didn’t previously indicate that these were melting endotherms. The manuscript has now been adjusted to add ‘the melting point’ on page 9 to clarify this.

  1. Figure 6 shows extremely large SDs; please elaborate on this issue and its relationship to the type of measurement.

Response: The data was acquired by measurement of the cleavage of a quenched fluorescent substrate over time. There are many other papers that use this assay (e.g. references Emergine Microbes Infect 2020;9:169, Biotechnol Lett 2016;38:1341) to investigate inhibition of sortase A. Most other groups do not provide error bars. This process is actually quite complex as it involves the release of drug from the protein nanoparticles, which may vary with respect to size and the different sites/forms of the protein aggregates at which the molecules bind, as well as the need for an enzymatic reaction to cleave the quenched substrate, and react this with tri-glycine substrate, which in turn may still represent a substrate for the enzyme, allowing some cycling to occur. In addition, there maybe precipitation that occurs and the potential for the protein or sortase A inhibitor to fluoresce at the excitation wavelength or to quench the emitted fluorescence. There is also the potential for variance between groups due to the differences in timing between when the enzyme is pipetted into each well. We incorporated groups to help control for these effects e.g. groups with all components except enzyme were used to control for background fluorescence, and groups without the inhibitor protein nanoparticles were used to measure that background reaction rate. Despite these errors, the data provides a useful tool for comparing sortase A inhibition between different compounds and concentrations. We have investigated inhibitors with much lower potency (e.g. peptidic molecules) and can clearly see the differences in inhibition between these and the inhibitors reported herein by this assay.

Reviewer 3 Report

molecules-2184800, Sortase A inhibitor (SrtAI)-loaded protein nanoparticles and their combinations with antimicrobial peptides show synergy against bacterial infections

The manuscript presents a good research, but it needs some important improvements. It needs a better structure and an objective view on the problem. Please don't try to present the data in a "manipulative" way that would confuse the readers.

The title should not contain abbreviations. Just remove (SrtAI). Also, for me the title sounds a little complicated and even strange for standard English.

In the abstract the authors should remove [I5, R8] mastoparan, and add maybe call it a mastopan derivative. In the main article, they should explain what [I5, R8] mastoparan means.

Row 46, detail on the sortase A importance as antivirulence target. See the following articles: Targeting Bacterial Sortases in Search of Anti-Virulence Therapies with Low Risk of Resistance Development. Pharmaceuticals, 2021

The section 76 to 91 seems to me out of place and redundant. It could be removed.

Row 180 and some other places: the bacterial species should be italics.

Row 277, detail on the methods section how the particle size distribution (Z-Average, PDI, and intensity mean hydrodynamic diameter) and surface charge (Zeta-potential) were measured

Figures 4 and 5 should be moved as supplementary material.

In figure 6 add a measure of error (like the confidence interval) for the presented average values. The points on the graph seem to have a large wide of distribution and the SD seem to be high.

The authors should be more careful in their conclusions. The inhibition of SrtA does not kill the bacteria. For a good anti-virulence agent, the MIC values should be high, not low. Also, the tested nanoparticles have a significant toxicity at higher doses.

On row 507: “could provide an alternate treatment for this medically important antimicrobial-resistant bacteria”. I don’t agree with this. The MIC values are low indicating a high pressure for the developing of resistant strains. The authors should have chosen compounds that inhibit SrtA without killing the bacteria, otherwise it seems pointless to talk about the antivirulence strategy. See : Sortase A (SrtA) inhibitors as an alternative treatment for superbug infections. Drug Discov Today. 2021 Sep;26(9):2164-2172

The authors should explain the logic in choosing the 4 substances. Just because they have a low solubility?  If sortase is considered the drug target, why the research uses also Gram negative bacteria that do not have this enzyme?

There are many editing mistakes. The authors should check the journal’s style and correct them.

Author Response

Reviewer 3 

Comments and Suggestions for Authors

  • The manuscript presents a good research, but it needs some important improvements. It needs a better structure and an objective view on the problem. Please don't try to present the data in a "manipulative" way that would confuse the readers. 

Response: The authors have formatted the manuscript in accordance with the Molecules Instructions for Authors ”The structure should include an Abstract, Keywords, Introduction, Materials and Methods, Results, Discussion, and Conclusions (optional) sections, with a suggested minimum word count of 4000 words”. Combined Results and Discussion sections, as used, are accepted in molecules and are seen in many recent publications. The results section delivers a concise and accurate explanation of the experimental results, along with their explanation. The document is objective, it indicates that there are issues associated with both the sortase A inhibitors and antimicrobial peptides used in the manuscript such as poor solubility, toxicity and cost; that combinations of these components are one of many options to help to develop new antimicrobials to treat superbug infections; and clearly indicates that further development of these platforms would be necessary to progress these systems towards the clinic. We are not sure what is exactly being requested with these comments.

  • The title should not contain abbreviations. Just remove (SrtAI). Also, for me the title sounds a little complicated and even strange for standard English.

Response: The abbreviation SrtAI has been removed from the title and the title changed to “sortase A inhibitor protein nanoparticle formulations demonstrate antibacterial synergy when combined with antimicrobial peptides” to improve its readability.

  • In the abstract the authors should remove [I5, R8] mastoparan, and add maybe call it a mastopan derivative. In the main article, they should explain what [I5, R8] mastoparan means.

Response:

As requested by the reviewer, the description [I5, R8] has been removed from the abstract, and the sentence changed to specify ‘a mastoparan derivative’.

The final paragraph on page 2 has been adjusted to indicate what [I5, R8] mastoparan means:

‘…and [I5, R8] mastoparan (MASTO; contains an isoleucine and arginine at positions 5 and 8, respectively, which ameliorates its toxicity compared to wildtype mastoparan).’

  • Row 46, detail on the sortase A importance as antivirulence target. See the following articles: Targeting Bacterial Sortases in Search of Anti-Virulence Therapies with Low Risk of Resistance Development. Pharmaceuticals, 2021

Response: To address this comment, the reference Pharmaceuticals 2021, 14(5), 415; https://doi.org/10.3390/ph14050415 that was provided by the reviewer has been included as reference 6 in the first paragraph of page 2.

  • The section 76 to 91 seems to me out of place and redundant. It could be removed.

Response:  The authors believe that this section is important for the following reasons:

  1. It prepares readers for the structure of the paper i.e., what will be presented in the sections below, and
  2. It describes the processes used to produce the protein nanoparticles, including links to a clear scheme (scheme 1), which illustrates how these are made. Molecules has a wide readership and scope. Knowledge of what protein nanoparticles are, and how they are made is not widely known, and thus this quick overview will be important for readers.

  • Row 180 and some other places: the bacterial species should be italics.

Response:  To address this comment, the names of bacterial species have been checked for italicisation, and italicised in Section 2.6, as well as the Figure 4 and Table 2 captions, where italics were missing.

  • Row 277, detail on the methods section how the particle size distribution (Z-Average, PDI, and intensity mean hydrodynamic diameter) and surface charge (Zeta-potential) were measured

Response: These were all measured by dynamic light scattering (DLS) as mentioned in section 2.4. They were measured on a Malvern Zetasizer Nano ZS as mentioned in section 2.2. To further address this comment, we have added to section 2.4 (page 4) that in addition to the samples being measured at 1 mg/mL in PBS, they were measured in disposable folded capillary cells, and measurements performed at 25 ℃. The particle sizes were also measured by transmission electron microscopy (TEM) as described in section 2.4.

  • Figures 4 and 5 should be moved as supplementary material.

Response: These figures could be moved to the supplementary material, however, they provide a useful visual colorimetric demonstration of the results of the broth microdilution and checkerboard experiments, which helps to illustrate how these were performed for readers, and allows for quick and simple visual comparison of data between groups.

Figure 4 provides a compact, easy to compare visual readout of the antimicrobial activities of each sortase A inhibitor protein nanoparticle against the four bacterial strains in our library.

In comparison, Figure 5 only shows checkerboard synergy data for the leading formulation (TC protein nanoparticles combined with pexiganan) against each of the bacterial strains in the library. The other data (in particular the curcumin and quercetin protein nanoparticles combined with antimicrobial peptides) have been placed in the supplementary materials so they can be viewed by interested readers (Figures S4-S6), whilst not taking up valuable space in the manuscript.

Based on these comments, the authors would like to keep figures 4 and 5 in the manuscript.

  • In figure 6 add a measure of error (like the confidence interval) for the presented average values. The points on the graph seem to have a large wide of distribution and the SD seem to be high.The authors should be more careful in their conclusions. The inhibition of SrtA does not kill the bacteria. For a good anti-virulence agent, the MIC values should be high, not low. Also, the tested nanoparticles have a significant toxicity at higher doses.

Response:  We agree with the authors comments that for a ‘good anti-virulence agent, the MIC values should be high, not low’. The have adjusted the last conclusion section ‘This data suggests these SrtAI formulations could be used as broad spectrum antimicrobials on their own’ to remove SrtAI from this sentence. ‘There is a measure of error on the graph (SD) already. We note that the measured IC50 values for each sortase A inhibitor protein nanoparticle were within the same order of magnitude, and so there was not a significant difference with respect to their inhibition of sortase A. This experiment was used mainly to compare against previous data where sortase A inhibition was measured in the presence of a significant concentration of DMSO (5% v/v), which was needed to keep the sortase A inhibitors dissolved when they weren’t formulated. We wanted to confirm that the presence of DMSO, which could have significant effects on protein structure and activity, did not affect the activity of these compounds. Comparing these data, it would appear that the IC50 data is similar for the protein nanoparticle formulations when compared to the DMSO formulations. The comments in this section are all simply what was observed in the experiment, and because there wasn’t a large difference between groups, we haven’t used this data as a parameter for selecting one formulation over another for further studies.

  • On row 507: “could provide an alternate treatment for this medically important antimicrobial-resistant bacteria”. I don’t agree with this. The MIC values are low indicating a high pressure for the developing of resistant strains. The authors should have chosen compounds that inhibit SrtA without killing the bacteria, otherwise it seems pointless to talk about the antivirulence strategy. See : Sortase A (SrtA) inhibitors as an alternative treatment for superbug infections. Drug Discov Today. 2021 Sep;26(9):2164-2172

Response:  We agree with the reviewers comment and have removed the sentence ‘This data suggests that these formulations could be used as broad-spectrum antimicrobials on their own, and owing to their efficacy against MRSA, could provide an alternate treatment for this medically important antimicrobial-resistant bacteria’. The location of this sentence suggests that we are referring to the SrtA inhibitor protein nanoparticle formulations alone, which was not our intent. We agree that these are not likely to be effective as an antimicrobial approach on their own.

  • The authors should explain the logic in choosing the 4 substances. Just because they have a low solubility?  If sortase is considered the drug target, why the research uses also Gram negative bacteria that do not have this enzyme?

Response:  The logic was not because they have low solubility. It would be preferable that the compounds possessed an appropriate hydrophilicity to allow them to be readily soluble under aqueous conditions. The reasons for selecting these compounds were because they are inexpensive, generally regarded as safe (GRAS) compounds, that possess sortase A inhibition activity, and are widely studied in the literature. We had conducted a cost analysis of literature reported sortase A inhibitors and noted that many were too expensive to be used as part of a widespread antimicrobial approach. It was hoped that formulations of these compounds could provide an inexpensive, and readily available way to improve the potency of antimicrobial peptides, so that their dose could be reduced, which in turn would also be expected to reduce their toxicity. This could provide a means to progress some antimicrobial peptides towards clinical use.

On re-reading this manuscript, we note that the reasoning for using these compounds is only briefly mentioned in the abstract ‘safe and inexpensive’, and is missing from the introduction. As such, we have added the following sentence on page 2, paragraph 3:

‘These SrtAIs were selected as they are readily available, inexpensive, safe, and widely used as dietary supplements’

We have also assessed these formulations against Gram-negative bacteria, as they provide some antimicrobial activity, albeit weak as you note in question 10 (and demonstrated for trans-chalcone and berberine in Figure 4), and so were interested to see if they might provide synergy against Gram-negative bacteria as well when combined with antimicrobial peptides. We selected strains that are leading causes of healthcare-related infections globally (E. coli and P. aeruginosa; ESKAPE pathogens). Of interest, only trans-chalcone showed synergy against the Gram-negative bacteria when combined with the antimicrobial peptide pexiganan.

  • There are many editing mistakes.The authors should check the journal’s style and correct them. 

Response: To address this comment, throrough proofreading and editing has been performed.

Round 2

Reviewer 3 Report

The authors responded to most of the comments and recommendations and improved the quality of their manuscript. Still, I must insist that figures 4 and 5 to be moved in the supplementary section. If any reader wants to see them, they are just one click away. The authors provided some answers to my questions in their response, but this information should be also available to the readers. Please integrate these responses in your final paper as for the readers to better understand the rational design of the research.

I'm sorry, but I forgot in my first review. Please, use colors in figure 6, 7 and 8. It would clearly improve the quality of the paper. It seems that the authors forgot to answer the question about the measure of error (like the confidence interval) for the presented average values. The points on the graph seem to have a large wide of distribution and the SD seem to be high. Please comment on these in your manuscript.

Author Response

Responses to Reviewer 3 – Round 2

  • The authors responded to most of the comments and recommendations and improved the quality of their manuscript. Still, I must insist that figures 4 and 5 to be moved in the supplementary section. If any reader wants to see them, they are just one click away.

Response: We have removed these figures and placed them in the supporting information and updated the figure numbering.

  • The authors provided some answers to my questions in their response, but this information should be also available to the readers. Please integrate these responses in your final paper as for the readers to better understand the rational design of the research.

Response: The reasons for using the compounds in this manuscript was previously added “These SrtAIs were selected as they are readily available, inexpensive, safe, and widely used as dietary supplements”.

We have added to the comments on Gram-negative bacteria by adding the following to the results section 3.3 “Since the Gram-negative strains do not express SrtA, this indicates that these compounds possess antimicrobial activities that are effective against these reference strains independent of SrtA inhibition activity”.

Additional detail on the measurement of particle size distribution and surface charge was previously added to the manuscript as requested.

The comments around adding confidence intervals to the IC50 data and discussing this has been dealt with in comment 4 below.

The discussion about the use of the IC50 experiment to compare against the activity of the same compounds when administered in DMSO was discussed in section 3.5 – e.g. DMSO at the necessary concentrations to keep these compounds dissolved “is likely to affect enzyme structure, selectivity and activity, with significantly lower concentrations (e.g., 0.1% v/v) usually preferred. Because the PNP formulations of each SrtAI can be directly dispersed in aqueous buffers, without the need for organic solvents to improve their solubility, an aim of this work was to assess how the inhibition of SrtA with these formulations compared to the data previously acquired in the presence of 5% (v/v) DMSO”.

  • I'm sorry, but I forgot in my first review. Please, use colors in figure 6, 7 and 8. It would clearly improve the quality of the paper.

Response: Have updated the graphs to colorize the bars.

  • It seems that the authors forgot to answer the question about the measure of error (like the confidence interval) for the presented average values. The points on the graph seem to have a large wide of distribution and the SD seem to be high. Please comment on these in your manuscript.

Response: We sought advice on this from a pharmacologist, who indicated that there was an issue with how the normalisation process that was used in Prism. This has been fixed, which has reduced the error, but also changed the IC50 values slightly, albeit the trend is still the same, and the IC50 values are not significantly different. We have updated the manuscript with this new graph, adjusted its figure number to Figure 4 (due to the removal of Figures 4 and 5 from the manuscript), colored the mean data points, and added the calculated 95% confidence intervals. As requested, we have commented on the 95% confidence intervals by indicating that these overlap for the BR, CUR and TC PNP groups, and therefore their IC50 values are essentially the same, while the QC PNPs were less potent as previously described (see page 12).